# Meeting Service Members Where They Are: Supporting Vegetable Consumption Through Convenient Meal Kits

**DOI:** 10.3390/nu17132136

**Published:** 2025-06-27

**Authors:** Saachi Khurana, Jonathan M. Scott, Christopher R. D’Adamo

**Affiliations:** 1Consortium for Health and Military Performance, Department of Military and Emergency Medicine, F. Edward Hébert School of Medicine, Uniformed Services University, Bethesda, MD 20814, USA; saachi.khurana.ctr@usuhs.edu (S.K.); jonathan.scott@usuhs.edu (J.M.S.); 2Henry M. Jackson Foundation for the Advancement of Military Medicine, Inc., Bethesda, MD 20817, USA; 3Department of Family & Community Medicine, University of Maryland School of Medicine, Baltimore, MD 21201, USA

**Keywords:** vegetable intake, military, nutrition, herbs, spices, readiness, convenience

## Abstract

Vegetable intake among military Service Members (SMs) is well below public health guidelines, with only 12.9% meeting the Dietary Guidelines for Americans (DGAs). Low vegetable consumption negatively impacts diet quality as measured by the Healthy Eating Index (HEI), and poses risks to health and performance. Given the high physical and mental demands of military life, improving diet quality, including through increased vegetable intake, is crucial for optimizing health and readiness. Providing meal kits may help improve vegetable intake by reducing access-related barriers for SMs living or working on a military base. Furthermore, the addition of spices and herbs is a readily modifiable accompanying approach to address taste-related barriers and increase intake that has shown promise in other populations with poor diet quality. **Background/Objectives**: This study aimed to evaluate whether heat-and-serve meal kits with spices and herbs could increase vegetable intake and liking among active-duty SM by simultaneously targeting barriers to healthy eating and modifiable sensory factors. **Methods**: Conducted at Naval Support Activity Bethesda, the study randomly distributed heat-and-serve meal kits (*n* = 400) featuring either spiced (*n* = 200) or plain versions (*n* = 200) of four vegetables (broccoli, carrots, cauliflower, and kale). Each kit contained a quick response (QR) code for participants to upload post-consumption photos and rate vegetable liking on a nine-point Likert scale. Food photography (SmartIntake^®^) was used to estimate vegetable consumption. Paired t-tests were used to determine differences between the intake of plain and spiced vegetables. **Results**: Intake of the heat-and-serve vegetables was very high for both the spiced and plain preparations (1.73 out of 2 cups, 87%). There was minimal difference (*p* = 0.87) between the consumption of spiced (1.75 cups) and plain (1.725 cups) vegetables, suggesting that both were well accepted. Overall, convenient and accessible meal options, alongside sensory-driven strategies, appear to improve some barriers to vegetable consumption in SM populations. **Conclusions**: Future studies should explore long-term outcomes and adaptability across different military environments, while considering additional factors, including convenience and time constraints, that influence dietary choices in the military.

## 1. Introduction

Addressing the gap in vegetable intake among military service members (SMs) is essential for improving diet quality and reducing the risk of chronic diseases and other health-related conditions. SM adherence to federal guidelines, using the Healthy Eating Index (HEI), is generally suboptimal. With only 12.9% of active duty personnel meeting the Dietary Guidelines for Americans (DGAs) for vegetable intake, it is clear that new approaches are needed to support optimal health and readiness [1,2,3,4]. Given the demanding physical and psychological requirements of military life, proper nutrition plays a critical role in enhancing both performance and resilience.

The military food environment on and around bases presents unique challenges to healthy eating for SMs. Prior studies have identified barriers to adopting healthier eating habits, including limited access to nutritious foods, high costs, military culture, stress, peer pressure, lack of access to food preparation, and the prevalence of more convenient fast food options [5,6]. To improve diet quality and specifically address the low vegetable consumption among SM, more innovative and targeted strategies are required. Initiatives targeting appropriated funded military dining facilities, including dietary interventions, menu modifications, and strategic food placement, show promise, as evidenced by a 10-point increase in HEI scores and improved vegetable consumption [7]. A study among U.S. Army soldiers demonstrated that these interventions also led to higher satisfaction with food flavor, highlighting the importance of taste in encouraging healthier eating and potentially overcoming barriers to vegetable intake [8].

Taken together, a plausible solution to increase vegetable consumption is targeting sensory properties by incorporating spices and herbs to enhance the flavor of vegetables. Previous studies in diverse settings, including urban high schools [9,10,11], rural communities [12], and universities [13,14,15], have consistently demonstrated that improving the taste and other sensory properties of vegetables through the use of spices and herbs can effectively promote healthier eating behaviors in populations with poor diet quality. Given that the majority of the military population is aged 18–25, these findings highlight the potential for implementing similar strategies in military environments to improve vegetable consumption.

To explore the potential of a convenience- and sensory-oriented intervention using spices and herbs in heat-and-serve meal kits to increase vegetable consumption in military settings, the initial phase of this research focused on identifying modifiable behaviors related to vegetable intake. In brief, the initial monadic sensory testing was used to determine whether vegetables enhanced with spices and herbs were preferred over identical vegetables without flavor enhancements (same amounts of fat and salt). The previous sensory testing revealed that vegetables with spices and herbs were rated significantly higher in overall appeal, flavor, and aroma compared to typical vegetable preparations (*p* < 0.03). Further findings identified barriers to vegetable intake, including appearance (42.9%), preparation style (41.3%), and taste (39.7%) [16]. Collectively, by focusing on modifiable factors, including easy access, appearance, taste, and preparation style, the military can implement practical solutions to create a more supportive food environment that promotes healthier eating behaviors. Therefore, the purpose of this study was to evaluate whether the addition of spices and herbs (spiced) to vegetables provided as part of heat-and-serve meal kits served to active-duty SMs will increase vegetable intake (primary outcome) and vegetable liking (secondary outcome) compared to plain vegetables. We hypothesized that vegetable consumption and acceptability would be high for both preparations due to the heat-and-serve convenience, but greater for spiced vegetables compared to plain vegetables.

## 2. Materials and Methods

### 2.1. Study Design

A multi-phase, comparative sensory assessment and feeding study was conducted among a sample of SMs on a military base in the United States. The study was conducted in two phases. Phase I involved a comprehensive sensory evaluation—comparing assessments of taste, appearance, aroma, and texture—between typical vegetable preparations (butter and salt) and spiced vegetable preparations (same amount of butter and salt with the addition of spices and herbs), utilizing a combination of Likert scale liking measurements and questionnaires [16]. In brief, while both preparations were relatively well-liked, the spiced preparations of most vegetables received higher liking scores than the typical versions. Qualitative analysis of open-ended feedback revealed a desire for more spices and herbs to be used in the spiced preparations. Accordingly, a spice and herb content increase of approximately 25% was added to the recipes evaluated in Phase I. Subsequently, Phase II, as described in this manuscript, focused on evaluating vegetable intake and was designed based on the outcomes of Phase I. The study received expedited review and was determined to be exempt from human subjects research by the Institutional Review Boards of the Uniformed Services University (approval No: USUHS.2022-105, 5 July 2022) and the University of Maryland School of Medicine (approval No.: HP-00095407, 3 January 2022). Written informed consent was waived by the Institutional Review Boards that approved the study because of the minimal risk of the intervention and the de-identified nature of all data that were collected. Instead, study volunteers were provided with an information sheet with study details and provided verbal consent to participate. The study was registered on ClinicalTrials.gov (NCT05499858).

### 2.2. Study Setting

The study was conducted at Naval Support Activity Bethesda (NSAB), a military base located in Bethesda, Maryland, that includes the Walter Reed National Military Medical Center, the Uniformed Services University (USU), and the Warrior Transition Brigade, along with more than 40 other tenant commands. The meal distribution and data collection were performed primarily at the student lounge located on the USU campus.

### 2.3. Study Sample

Study volunteers were required to be at least 18 years of age, in the military with an Enlisted rank of E1–E4 or an Officer rank of O1–O3, and able to read and write in English. Study participants were recruited throughout NSAB via word of mouth, printed flyers, and email communication with key leaders and other stakeholders.

### 2.4. Intervention

Phase I focused on assessing modifiable barriers to vegetable intake among active-duty SMs through a combination of questionnaires, focus groups, and sensory testing. The results from Phase I provided insights into the acceptable spices, herbs, and vegetable recipes for this population, informing the design of meal kits and the intervention approach for Phase II [16].

In Phase II, the intervention focused on comparing vegetable intake and liking with and without the use of spices and herbs among active-duty SMs. Drawing on the findings of Phase I, meal kits were strategically designed and distributed to participants. Meal kits were prepared by a local catering company (Rouge Fine Catering, Hunt Valley, MD, USA) and contained vegetables (two cups) either prepared with spices and herbs (“spiced”) or without spices and herbs (“plain”), alongside a consistent protein (5 oz grilled chicken breast) and high starch food (½ cup mashed potatoes) for all meals under study. This consistency of the other components of the meals ensured uniformity across meals and minimized the potential for confounding by differing accompanying meal components. The vegetables analyzed in the sensory testing process in Phase I and included in the meal kits in Phase II were broccoli, carrots, cauliflower, and kale. The “plain” vegetable recipes included added salt and compound butter. In addition to the same amounts of salt and compound butter, flavor profiles of the “spiced” vegetables were savory broccoli (most abundant spices and herbs—garlic powder, onion powder, black pepper, and mustard seed), curry cauliflower (most abundant spices and herbs—garlic powder, onion powder, turmeric, and coriander), sweet spice carrots (most abundant spices and herbs—vanilla and cinnamon), and smoky kale (most abundant spices and herbs—garlic powder, onion powder, paprika, and oregano).

Across nine sessions, approximately 400 meal kits were distributed, featuring either the “spiced” or “plain” version of one of the four vegetables. Only one version—either spiced or plain—was distributed per session. The version (spiced or plain) of vegetables served at each of the nine sessions was determined through randomization. This distribution process resulted in 400 meals distributed in total, 100 meals per vegetable, with 50 spiced and 50 plain. Meals were made available between 11:00 a.m. and 1:00 p.m. in the student lounge to accommodate participants’ busy schedules. Participants could either receive a pre-heated meal for immediate consumption or select a frozen meal kit to take home, providing flexibility to suit their needs.

### 2.5. Outcomes

Each meal kit included a quick response (QR) code that directed participants to a SurveyMonkey page, allowing them to anonymously upload photos of their completed meals and rate their liking of the vegetables using a 9-point Likert scale, where 1 represented “dislike extremely” and 9 indicated “like extremely.” Data were collected to evaluate both the quantity of vegetables consumed and the influence of spices and herbs on participants’ vegetable preferences. To accurately measure vegetable intake, SmartIntake^®^ Technology (Pennington Biomedical Research Center, Louisiana State University, Baton Rouge, LA, USA), a form of Remote Food Photography Method (RFPM), was utilized [17]. In brief, SMs used a mobile device to capture images of their meals post-consumption, utilizing photo calibration cards to ensure precise volumetric assessments of the vegetables consumed. These post-consumption photos were then compared to the pre-consumption images taken by the research team before meal distribution, allowing for a volumetric and accurate assessment of vegetable intake. In brief, this specific RFPM technology demonstrated greater than 93% accuracy in the original validation publication [17]. More generally, many publications on RFPM have demonstrated high accuracy and reliability in quantifying food intake in a variety of populations [17,18,19], and specifically among military populations [20].

### 2.6. Statistical Analysis

Comparison of vegetable intake between plain vegetable recipes and spiced vegetables was performed using a *t*-test. Pooled and Satterthwaite variance methods were used, and no meaningful differences were determined. Thus, pooled (equal) variance assumptions were used throughout these analyses. Statistical significance was defined as *p* < 0.05. All statistical analyses were conducted in SAS Version 9.4.1 (Cary, NC, USA).

## 3. Results

The preliminary phase of this research identified appearance, preparation style, and taste as the main barriers to vegetable intake in military dining facilities, with sensory testing showing a preference for spiced vegetables over traditional preparations in terms of appeal, flavor, and aroma (*p* < 0.03) [16]. Building upon the findings from Phase I, Phase II further explored these barriers, distributing a total of 400 meals to gather data on vegetable consumption and liking ratings for both spiced and plain preparations of four vegetables: broccoli, carrots, cauliflower, and kale. Fifty meals were distributed per vegetable. Complete photographic data were collected for analysis from 40 participants for spiced broccoli and 41 for plain broccoli. For carrots, 31 photo sets were received for spiced and 36 for plain. Spiced cauliflower and plain cauliflower each had datasets featuring photos from 41 and 40 participants, respectively. Finally, 38 sets were collected for spiced kale and 37 for plain kale. A total of 304 participant datasets were analyzed. Mean [standard deviation] liking scores for each vegetable are presented in Table 1.

Spiced preparations of broccoli, cauliflower, and kale received higher liking ratings than their plain counterparts, with broccoli (mean = 7.34) and kale (mean = 7.42) having the highest ratings. However, for carrots, the plain preparation was rated higher (mean = 6.33) compared to the spiced (mean = 6.16). Overall, the differences in liking ratings between spiced and plain preparations were minimal; the largest difference was observed for kale (difference = 0.37), but it was not statistically significant (*p* ≥ 0.26).

Vegetable consumption was measured using the food photography method, which quantified the amount of each two-cup vegetable serving consumed by estimating the cups of vegetables remaining. The mean number of cups consumed for both spiced and plain preparations is presented in Table 2. Intake of both varieties of all vegetables was high, with intake ranging from 1.5 to 1.9 out of 2.0 cups. Mean intake for spiced preparations was 1.75 cups, and 1.73 for plain preparations (*p* = 0.87). For broccoli, carrots, and kale, participants consumed slightly more spiced vegetables than plain. In contrast, cauliflower was consumed in greater amounts in the plain preparation than the spiced version. The intake differences were all modest and none reached statistical significance (*p* ≥ 0.1).

## 4. Discussion

### 4.1. Acceptability, Accessibility, and Availability

This study aimed to evaluate whether heat-and-serve meal kits with spices and herbs could increase vegetable intake and liking among active-duty SMs, by simultaneously targeting barriers to healthy eating and modifiable sensory factors. Although similar research has been completed with high school students [9,11], this study is among the first to address modifiable barriers to vegetable consumption in a military population by incorporating meal kits with heat-and-serve vegetables with spices and herbs. The results revealed small, insignificant differences between spiced and plain vegetables, with overall high likeability and intake across both options. This may be attributed to the fact that all of the meals were chef-inspired, prepared by a professional catering company, and informed by consumer preferences, ensuring that both versions of the meals were flavorful and appealing. The incorporation of spices and herbs likely played a moderating role, enhancing the sensory experience without overwhelming the natural taste of the vegetables [16]. By focusing on creating meals that were both nutritious and convenient, the development process ensured that the dishes would cater to a wide range of palates, increasing the likelihood of consumption. Although greater differences between the spiced and plain vegetables were anticipated, the similar likeability and consumption scores suggest that the meal design successfully promoted vegetable intake. The minimal variation in these scores highlights the effectiveness of creating meals that prioritize both taste and nutrition.

Delivering these interventions “where service members are”—in their daily environments, such as where they live, work, and/or train—is a critical factor in ensuring success. The convenience of offering healthy, flavorful vegetable options in familiar and frequently used settings addresses a significant barrier to healthy eating by reducing the time, effort, and decision-making required to choose nutritious options [7,8]. Convenience plays a vital role in shaping food choices, particularly in military environments where SMs often face time constraints and demanding schedules [5,16]. Individuals tend to opt for the most convenient food choices, which are often low in nutrient density. Although convenience is a critical factor, there is limited research exploring meal kits as a strategy to improve accessibility and nutrient intake among SMs. This gap highlights the value and potential significance of integrating meal kits into their daily environments.

By incorporating vegetables enhanced with spices and herbs directly into meal kits provided in common areas, this intervention can be seamlessly integrated into daily routines, enhancing convenience, vegetable intake, and overall dietary quality. This approach minimizes additional preparation or effort, promoting healthier food choices if offered. Microwavable meals are especially convenient in settings where time and resources are limited, including military environments. With microwave ovens widely available in housing and workspaces, SMs can quickly prepare a hot, nutritious meal with minimal effort. Additionally, SMs have access freezer space, allowing them to store meals and grab a quick, nutritious option when needed. This convenience allows them to fit healthy eating into their fast-paced routines without sacrificing time for food preparation or cleanup. Additionally, making these options readily available within the settings where SMs live and work reduces their reliance on fast food options, which often lack nutritious choices. Overall, these meal kits enhance the availability and accessibility of nutrient-dense meals, including vegetables.

### 4.2. Sustainability and Scalability

By using common, readily accessible spices and herbs to enhance the flavor of vegetables, this approach is not only low-cost but also adaptable to various dining environments across military installations. Whether in large-scale dining facilities (cafeterias), smaller barracks (housing), or kitchens, the intervention can be implemented with minimal changes to existing food service operations. Military installations vary in terms of size, structure, and available resources, but the approach of using spices and herbs to enhance flavor can be applied universally. In large-scale dining facilities, where hundreds or thousands of meals are served daily, this strategy can be integrated into bulk meal preparation processes without altering the overall workflow. In contrast, in smaller settings like barracks kitchens or mobile field kitchens, the intervention can be implemented on a smaller scale, using meal kits or pre-prepared food options.

The intervention’s flexibility is valuable in field operations or deployed settings, where the availability of fresh ingredients may be limited, and SMs often rely on pre-packaged meals such as Meals Ready-to-Eat (MREs). Spices and herbs can easily be incorporated into these meal kits, enhancing flavor without requiring perishable ingredients or additional cooking equipment. This adaptability ensures that SMs have access to flavorful, nutritious meals regardless of their location or mission, helping to maintain diet quality even in austere environments.

### 4.3. Future Directions and Potential Implications

Moving forward, several areas warrant further exploration. Future research should investigate how similar flavor-enhancing strategies can be adapted and scaled to diverse military dining environments, including dining facilities, barracks kitchens, and various locations on installations. By evaluating how these interventions can be integrated into both large-scale and small-scale meal preparation, researchers can identify best practices for ensuring widespread use across installations. Additionally, the potential integration of these strategies in deployed field operations, where fresh ingredients may be limited, could play a role in maintaining diet quality in austere environments. Given the success of the intervention in a training setting, expanding the application to military field operations and deployed settings could be a potential solution to improving diet quality in a resource-constrained environment. Investigating the potential for spices and herbs to improve the acceptability and likeability of pre-packaged meals stands to improve dietary quality and adherence to healthier eating habits during deployments. While the intervention demonstrated high acceptability and likeability in the short term, further research is needed to assess its impact on long-term dietary behavior and consumption patterns. By monitoring vegetable intake over extended periods in a larger sample size, researchers can evaluate the sustainability of convenient (microwavable), sensory-driven approaches to promoting consistent vegetable consumption and the impact of diet quality on health, readiness, and performance.

A critical aspect of successful implementation is involving key stakeholders, including military leadership, food service providers, and the SMs themselves. Stakeholder-informed interventions ensure that the solutions are practical, culturally relevant, and aligned with the needs and preferences of the population. Involving food service staff in menu development and sensory testing also ensures that the intervention is logistically feasible and can be sustained over time. Buy-in from military leadership is essential for ensuring the long-term success of the intervention. When leaders prioritize healthy eating as part of a broader commitment to improving SM well-being and operational readiness, it creates a culture that supports these behaviors [21].

Although this study focused on SMs, there may be opportunities to apply these findings to other populations with similarly structured environments, including colleges, hospitals, or other institutional dining settings. Investigating how convenience-oriented and sensory-driven interventions like this one can be adapted for different age groups, cultural backgrounds, and dietary needs could lead to broader applications of this approach. To ensure successful implementation and long-term sustainability, future research should deepen stakeholder involvement, especially from military leadership and food service staff. Understanding the logistical challenges and preferences from the perspective of those who prepare and serve the meals will help refine intervention strategies. Additionally, building leadership buy-in will be crucial for the allocation of resources and consistent promotion of healthy eating as a priority for SM well-being.

### 4.4. Strengths and Limitations

This study has several strengths, including directly addressing the barriers to vegetable intake among military SMs by leveraging spices and herbs to enhance flavor. This approach, focusing on sensory-driven strategies, is innovative and practical, as it enhances flavor while improving nutrition. The chef-inspired meals, informed by consumer preferences, ensured the dishes were both appealing and nutritious, which likely contributed to the high overall likeability across both spiced and plain vegetable options.

The involvement of key stakeholders, including military leadership and food service staff, is another strength. This collaboration ensures the intervention is practical, culturally relevant, and sustainable over time, increasing its chances of long-term success. The focus on providing convenient, nutrient-dense meal options directly in SMs’ daily environments is a strength, as it addresses many of the previously identified barriers to healthy eating.

Another strength of the study was its robust design, which included a phased implementation approach, standardized meal preparation and distribution, and consistent data collection procedures across all conditions. These elements minimized bias, improved internal validity, and supported reliable comparisons between spiced and plain vegetable options in an operational military setting.

This study also has several limitations that should be considered. First, the study’s sample size, although adequate for this research, is not generalizable to the broader active-duty SM population, particularly across service branches and regions. The study was conducted on a single military base (Naval Support Activity Bethesda), which limits generalizability to other military installations, particularly those in different geographic regions or with varying access to resources. Additionally, the intervention was short-term, with data collection focused on the immediate consumption and enjoyment of vegetables. While this provides valuable information about initial acceptability, it does not assess the long-term impact of the intervention on sustained vegetable consumption or overall dietary behavior. Further research is needed to determine whether adding spices and herbs can lead to lasting changes in eating habits among SMs. The study also relied on self-reported data and photos submitted by participants to assess vegetable intake. While the use of SmartIntake^®^ Technology is common in nutrition research [17,22,23] and improves the accuracy of intake measurement, representing one of this study’s strengths, the reliance on participant-uploaded data introduces potential issues, including incomplete or inconsistent submissions. Furthermore, because vegetable intake was its primary focus, the study did not consider potential shifts in the consumption of other food groups or overall caloric intake, which could provide additional insights into dietary changes. On a related note, the unexpectedly high intake of plain vegetables resulted in ceiling effects when comparing spiced and plain vegetables. Finally, the study’s design focused primarily on the flavor of vegetables as a modifiable factor of vegetable intake, but other important factors, such as convenience, cost, and availability of healthy options, were not explored in as much comparative depth. These factors are critical in military environments where time constraints and operational demands can heavily influence food choices. Future research should integrate and measure these aspects to provide a more holistic view of the challenges and potential solutions to improving diet quality among SM.

## 5. Conclusions

In summary, by focusing on acceptability, accessibility, and availability, as well as fostering stakeholder engagement, this intervention offers a promising strategy for improving vegetable consumption and overall diet quality in military settings. If expanded and tailored effectively, the intervention has the potential to positively impact the health, performance, and readiness of SMs. This approach aligns with broader efforts to create a military nutrition environment that supports long-term health and operational readiness. Future studies should explore long-term outcomes and adaptability across different military environments, while considering additional factors such as convenience and time constraints that influence dietary choices in the military.

## Figures and Tables

**Table 1 nutrients-17-02136-t001:** Mean likeability * scores for spiced and plain vegetable preparations.

Vegetable	Mean [SD]	*p*-Value
Broccoli—Spiced (*n* = 40)	7.3 [1.22]	0.74
Broccoli—Plain (*n* = 41)	7.3 [1.26]	
Carrots—Spiced (*n* = 31)	6.2 [1.53]	0.62
Carrots—Plain (*n* = 36)	6.3 [1.30]	
Cauliflower—Spiced (*n* = 41)	6.6 [1.67]	0.38
Cauliflower—Plain (*n* = 40)	6.3 [1.46]	
Kale—Spiced (*n* = 38)	7.4 [1.22]	0.26
Kale—Plain (*n* = 37)	7.1 [1.58]	

* Likeability scores range from 1–9, with higher scores indicating greater preference.

**Table 2 nutrients-17-02136-t002:** Mean cups consumed for spiced and plain vegetable preparations.

Vegetable	Cups Consumed (Mean [SD])	*p*-Value
Broccoli—Spiced	2.0 [0.01]	0.11
Broccoli—Plain	1.9 [0.37]	
Carrots—Spiced	1.6 [0.58]	0.61
Carrots—Plain	1.5 [0.62]	
Cauliflower—Spiced	1.5 [0.53]	0.10
Cauliflower—Plain	1.7 [0.50]	
Kale—Spiced	1.9 [0.26]	0.22
Kale—Plain	1.8 [0.54]	

## Data Availability

The original contributions presented in this study are included in the article. Further inquiries can be directed to the corresponding author.

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
