# Peer review of "Meeting Service Members Where They Are: Supporting Vegetable Consumption Through Convenient Meal Kits"

_nutrients, 2025, doi:10.3390/nu17132136_

Round 1
Reviewer 1 Report (Previous Reviewer 2)
Comments and Suggestions for Authors
This study is aimed to evaluate whether the addition of spices and herbs (spiced) to vegetables provided as part of a heat-and-serve meal kit meals served to active-duty service members will increase vegetable intake (primary outcome) and vegetable liking (secondary outcome) compared to plain vegetables.
This manuscript can not be accepted as an article (original scientific investigation). Maybe it can be a "short communication" or presentation of preliminary results.
The results show 2 tables with four vegetables (spicy and unspicy) and no significant differences were found for any of the data (amount of consumption (table 2) and spiciness (table 1).
This is followed by a discussion that is three times longer than the results (L205-L344 vs L173-L202) in which some extremely important segments (subchapters 4.3. and 4.4.) are discussed without any related references
I'm sorry, but this is not acceptable for an original scientific paper.
Author Response
Reviewer 1:
This study is aimed to evaluate whether the addition of spices and herbs (spiced) to vegetables provided as part of a heat-and-serve meal kit meals served to active-duty service members will increase vegetable intake (primary outcome) and vegetable liking (secondary outcome) compared to plain vegetables.
This manuscript can not be accepted as an article (original scientific investigation). Maybe it can be a "short communication" or presentation of preliminary results.
Response: Thank you for your feedback. We respectfully disagree with your assessment of our manuscript and defer to the Editor for a final determination as to which article type is most appropriate. The study presents the final results of a two-phase study; therefore, a presentation of preliminary results assumes additional results will follow up. Also, studies using a similar methodology and approach, referenced in this manuscript, have been published in Nutrients; therefore, we wish to be considered as an original scientific investigation under the Nutrition Methodology & Assessment Section.
The results show 2 tables with four vegetables (spicy and unspicy) and no significant differences were found for any of the data (amount of consumption (table 2) and spiciness (table 1).
This is followed by a discussion that is three times longer than the results (L205-L344 vs L173-L202) in which some extremely important segments (subchapters 4.3. and 4.4.) are discussed without any related references
Response: Thank you for your feedback. A lack of significant findings is an unfair critique and misses the core focus of the study, which is to exploring novel means to increase vegetable intake amongst a population that has historically had very low intake of vegetables. The discussion is framed around key aspects of improving vegetable intake and considerations for further research.
I'm sorry, but this is not acceptable for an original scientific paper.
Response: Thank you for your feedback. We respectfully disagree with your assessment of our manuscript and defer to the Editor for a final determination as to which article type is most appropriate.
Reviewer 2 Report (New Reviewer)
Comments and Suggestions for Authors
This is a good topic, with the rationale well explained as are the results. It is potentially very useful both for the reasoning behind it but also for the practical application.
I make the following suggestions - all very minor.
Line 42 Reference 1 – would be helpful to give better access to the paper by including https://apps.dtic.mil/sti/tr/pdf/ADA582287.pdf
Line 42 Reference 4 Better to cite exactly as given Nutrients. 2020 Dec 31;13(1):122. doi: 10.3390/nu13010122.
Line 48 – Excellent references. Suggest also including factors in ref 6 such as stress, peer pressure and lack of access to food preparation. This last one has particular relevance to your paper’s suggestion of ‘heat-and-serve’ packs.
Line 52 – a good reference. Slight error in your listing of it with ’ instead of ‘and’
Line 59 – The results are not a problem but might be useful to note in the reference list that references 13 and 14 were both funded by a company that markets seasonings.
Line 69 – as you are referring to what was found in ref 16 (which was done with 2 out of 3 in this new paper), it would be better to say Previous sensory testing…..
Line 128 – suggest you say ‘high starch food’ (mashed potatoes) or ‘starchy food’ (mashed potatoes)
Line 165 – suggest you add the word ‘and’ before specifically. This suggestion is because one of the previous references was for children and without the word ‘and’ it sounds as though all were in military populations.
Lines 189-190 (the table) – I would query the use of two decimal places here
Line 198 – I would query the use of 3 decimal places here
Line 278 – Use either ‘barrack kitchens’ or ‘barrack’s kitchens’
Author Response
Reviewer 2:
This is a good topic, with the rationale well explained as are the results. It is potentially very useful both for the reasoning behind it but also for the practical application.
I make the following suggestions - all very minor.
Line 42 Reference 1 – would be helpful to give better access to the paper by including https://apps.dtic.mil/sti/tr/pdf/ADA582287.pdf
Response: Thank you for your suggestion, we have added the link to the references.
Line 42 Reference 4 Better to cite exactly as given Nutrients. 2020 Dec 31;13(1):122. doi: 10.3390/nu13010122.
Response: Thank you for catching this, we have updated the reference as cited in Nutrients.
Line 48 – Excellent references. Suggest also including factors in ref 6 such as stress, peer pressure and lack of access to food preparation. This last one has particular relevance to your paper’s suggestion of ‘heat-and-serve’ packs.
Response: Thank you for your suggestion—these important factors have now been incorporated into the section. Line 48 now reads “Prior studies have identified barriers to adopting healthier eating habits including limited access to nutritious foods, high costs, military culture, stress, peer pressure, lack of access to food preparation, and the prevalence of more convenient fast food options.5,6”
Line 52 – a good reference. Slight error in your listing of it with ’ instead of ‘and’
Response: Thank you for catching this error. We have removed the “ ’”
Line 59 – The results are not a problem but might be useful to note in the reference list that references 13 and 14 were both funded by a company that markets seasonings.
Response: Thank you for this note. The McCormick Science Institute is a nonprofit organization, and funding disclosures are provided in the original research articles. Additionally, the seasonings used in those studies are not commercially marketed and were developed solely for research purposes. Since these studies are only referenced briefly and not examined in depth, we believe it is not necessary to include this detail in the manuscript.
Line 69 – as you are referring to what was found in ref 16 (which was done with 2 out of 3 in this new paper), it would be better to say Previous sensory testing…..
Response: Thank you for your recommendation, we have added “previous” and “the” to clarify. This section now reads “In brief, the initial monadic sensory testing was used to determine whether vegetables enhanced with spices and herbs were preferred over identical vegetables without flavor enhancements (same amounts of fat and salt). The previous sensory testing revealed that vegetables with spices and herbs were rated significantly higher in overall appeal, flavor, and aroma compared to typical vegetable preparations (P < .03).”
Line 128 – suggest you say ‘high starch food’ (mashed potatoes) or ‘starchy food’ (mashed potatoes)
Response: Thank you for your suggestion. We have added “high starch food” for clarification.
Line 165 – suggest you add the word ‘and’ before specifically. This suggestion is because one of the previous references was for children and without the word ‘and’ it sounds as though all were in military populations.
Response: We added the word “and” before specifically in line 165.
Lines 189-190 (the table) – I would query the use of two decimal places here
Response: Thank you for your suggestion, we have updated table 1 so use 1 decimal place.
Line 198 – I would query the use of 3 decimal places here
Response: Thank you for your suggestion, we have updated 1.725 to read 1.73.
Line 278 – Use either ‘barrack kitchens’ or ‘barrack’s kitchens’
Response: Thank you for your suggestion, we have used “barrack kitchens”
Reviewer 3 Report (New Reviewer)
Comments and Suggestions for Authors
Overall: I enjoyed reading the article. It was easy to read, of interest to the military community, and likely of interest to the nutrition community at large due to potential for scalability to the civilian sector. The design approach was clear and simple, which is a strength. As a reader I would like some information about the demographic characteristics of your sample and some clarity of how many were recruited, enrolled, and distributed to the tasting groups. Additionally, I struggled with the discussion as much of it appeared to extend beyond what your findings can say. That being said, all good information if reorganized to be clear to the reader that you are interpreting your findings in relation to Considerations (or Implications) for Practice and/or Future Investigation. I’ve also provided some suggestions to improve clarity (see attached). Thank you for the opportunity to review and I look forward to your revisions.
Line 140: A little confusing. Here you state nine sessions x ~50 kits is ~450 meals. But on lines 144-145 you state 400 meals of 50 spiced/50 plain - it seems as it that would be 8 sessions instead of 9. Please clarify.
Line 167 – Statistical Analysis: Were demographic statistics performed on the participants? It would benefit the reader to present something in the results about the mean age, % sex, % enlisted/officer, etc. And to know how many participated in total to clarify question from earlier. Was it: 50 enrolled who participated in all 9 sessions to taste-tested each of the vegetables spiced and plain (participants completed all sessions). 2) 100 enrolled who participated in 2 sessions with each one tasting the spiced and plan version of one vegetable (new participants for each vegetable). Or 3) something different?
Line 173: At minimum there should be a clear statement about how many participants were recruited, enrolled and participated by vegetable type.
Line 189: Table1 should be able to stand alone. It would benefit the reader if you added the Likability score range in the footnote along with a direction (i.e., higher scores representing increased liking of the food item).
Line 203: I struggled a bit with the discussion. One thing that seems to be missing is citing other research studies relevant to your findings on improving the flavor / acceptability of vegetables and promoting change to increase vegetable intake.
Perhaps start the discussion by restating your objective. It might be easier to see that a logical discussion, based upon your methods/results, would be to discuss acceptability (section 4.1) and strategies to increase vegetable intake (section 4.2), and then move onto Implications for Practice (section 4.3) and/or Future Investigation (section 4.4) where you discuss your thoughts on accessibility, availability, sustainability, scalability, and key stakeholders. Much of your discussion is supposition that extends beyond the means of your findings. But it certainly is relevant from an Implication (or Considerations) for Practice perspective.
Line 204: You study didn't assess accessibility or availability. Your objective was to observe in adding herbs and spice would lead to increased vegetable intake in SM.
Line 205: Do you have a citation for studies with high school students?
Line 208: Consider adding "insignificant" since even though small differences, they are not significant and could be different by chance.
Lines 212-213: Do you have a citation for this presumption? What about the consideration that many of your participants are likely working in the healthcare field NSA Bethesda is not like a regular military installation) and may be more likely to eat vegetables for health than the majority of other military occupations?
Lines 229-231: This seems more appropriate if moved up to the last paragraph (end of line 228).
Line 235: Can you say "making it more likely" if you had no significant increase in consumption. Consider instead "promoting healthier food choice..." or "enabling…".
Line 237: I'm not sure this is a valid statement. Military accommodations depend on the rank of the SM, installation, type of duty assignment, etc. Some SMs have mini-refrigerators in their rooms or share a refrigerator in a community kitchen in the barracks. Consider revising to remove "ample".
Lines 246-255: Was stakeholder engagement part of the prior paper focused on Phase 1? There is nothing describing the who and what of stakeholder engagement in the methods or results. Perhaps this discussion is really part of Implications for practice. Not a discussion of how your results fit into the current body of knowledge relative to your objective - to examine if herbs and spice will increase vegetable intake in SM.
Line 256: You didn’t examine sustainability or scalability – this really belongs with Considerations (or Implications) for Practice or within a new section for Recommendations for Military Stakeholders.
Line 269: Citation for MREs?
Lines 269-270: Is there a citation to confirm "can be easily incorporated". The US Army Natick Research, Development and Engineering Center is responsible for development and testing of MREs and other operational rations. They would likely be involved in development and likely not an easy incorporation.
Line 275: You could make this section solely focused towards future investigation if moving the accessibility, availability, sustainability, and stakeholder content to a Considerations for Practice section.
Line 289: Need to be careful with interpretation of results. You already had high acceptability and likability scores at baseline with no significant increase in intake with addition of spice and herbs.
Line 291: consider adding "in a larger sample size" after "periods".
Line 306: Typically, strength and limitations focus on important characteristics of the study design. One strength you are missing is the phased approach with consistency in meal prep., distribution, and data collection methods to control for bias.
Line 308: Did you collect data about the barriers? You can't really state it directly addressed the barriers to vegetable intake, unless citing studies that support that presupposition.
Line 314: Did you report on any of these interactions? Or was this part of Phase 1 cited elsewhere? You talk about the value of having stakeholder input in the section starting on line 246, but I really didn't see any of this highlighted in the methods or results.
Line 320: as mentioned earlier it is not clear to the reader what your actual sample size was and presented no demographic characteristics about the participants.
Line 322: This definitely limits generalizability. It is appropriate to remove "may". Also, consider that your population at NSAB is comprised of many SM working in the healthcare field, and may be more inclined to consume more vegetables, not to mention the social desirability of eating more vegetables when participating in a study.
Line 332: One could consider the SmartIntake Tech a strength compared to the most common data collection method of self-reported intake. Most tools have their strengths and limitations. You could consider including it as a strength as well.
Line 346-347: You just stated a limitation is that convenience, cost, and availability were not explored in as much depth (lines 339-340). And as mentioned earlier, you really only explored the acceptability of vegetables options and intake of plain vs spiced vegetables. Your methods and results do not support including accessibility, availability, or stakeholders in your summary - except to provide a recommendation for future examination. Instead consider focusing on the elevated vegetable intake regardless of spice / herbs, likely due to chef-inspired recipes and your small, specific population. And include that future research is warranted in a larger, more diverse sample of SM to examine if intake between plain and spiced versions was truly by chance considering the plain vegetable intake was higher than expected.
Author Response
Overall: I enjoyed reading the article. It was easy to read, of interest to the military community, and likely of interest to the nutrition community at large due to potential for scalability to the civilian sector. The design approach was clear and simple, which is a strength. As a reader I would like some information about the demographic characteristics of your sample and some clarity of how many were recruited, enrolled, and distributed to the tasting groups. Additionally, I struggled with the discussion as much of it appeared to extend beyond what your findings can say. That being said, all good information if reorganized to be clear to the reader that you are interpreting your findings in relation to Considerations (or Implications) for Practice and/or Future Investigation. I’ve also provided some suggestions to improve clarity. Thank you for the opportunity to review and I look forward to your revisions.
Response: Thank you for your thoughtful and constructive review. We sincerely appreciate your positive feedback regarding the clarity, simplicity, and relevance of our study design, as well as its potential applicability beyond the military context. We also appreciate your suggestions for improving the manuscript. In response to your request for additional clarity on sample characteristics and participation, we have revised the Methods and Results sections to include more detailed information on the number of participants recruited and how meals were distributed across sessions. Regarding your concerns about the Discussion section, we agree that some of the content extended beyond the immediate findings. We have reorganized parts of this section to help differentiate between interpretations grounded in the data and broader considerations for practice and future research. Thank you again for your valuable feedback and for the opportunity to improve our manuscript.
Line 140: A little confusing. Here you state nine sessions x ~50 kits is ~450 meals. But on lines 144-145 you state 400 meals of 50 spiced/50 plain - it seems as it that would be 8 sessions instead of 9. Please clarify.
Response: Thank you for catching this error. We distributed a total of 400 meals over 9 sessions. The goal was 50 meals per session, in 8 sessions. However, we added another session to finish out the 400 meals. We have adjusted the wording to read “Across nine sessions, approximately 400 meal kits were distributed, featuring either the "spiced" or "plain" version of one of the four vegetables”.
Line 167 – Statistical Analysis: Were demographic statistics performed on the participants? It would benefit the reader to present something in the results about the mean age, % sex, % enlisted/officer, etc. And to know how many participated in total to clarify question from earlier. Was it: 50 enrolled who participated in all 9 sessions to taste-tested each of the vegetables spiced and plain (participants completed all sessions). 2) 100 enrolled who participated in 2 sessions with each one tasting the spiced and plan version of one vegetable (new participants for each vegetable). Or 3) something different?
Response: Thank you for your comment. To maintain participant anonymity and minimize response burden, we used a brief, two-item post-meal survey and did not collect detailed demographic information such as age, sex, or rank. However, we verified that all participants met the eligibility criteria prior to participation. Regarding participation, the study design most closely aligns with option 3: approximately 50 participants were recruited for each vegetable preparation. Participants were invited to attend multiple meal sessions, with each of the eight primary sessions featuring a different vegetable (spiced or plain versions). A ninth session was added to accommodate logistical constraints that prevented full meal distribution during an earlier session. To summarize, we planned eight sessions, each distributing 50 meals of a specific vegetable preparation (spiced or plain). A ninth session was added to complete distribution of a vegetable that could not be fully distributed during its originally scheduled session.
Line 173: At minimum there should be a clear statement about how many participants were recruited, enrolled and participated by vegetable type.
Response: Thank you for your comment, to clearly address participation we have added the following statement. “All 50 meals were distributed per vegetable. Complete photographic data was collected for analysis from 40 participants for spiced broccoli and 41 for plain broccoli. For carrots, 31 photo sets were received for spiced and 36 for plain. Spiced cauliflower and plain cauliflower each had 41 and 40 participant data sets respectively. Finally, 38 sets were collected for spiced kale and 37 for plain kale. A total of 304 participant data sets were analyzed.” Please see lines 182-187.
Line 189: Table1 should be able to stand alone. It would benefit the reader if you added the Likability score range in the footnote along with a direction (i.e., higher scores representing increased liking of the food item).
Response: Thank you for this suggestion, we have included a footnote in Table 1 to guide readers. The footnote reads “Likeability scores range from 1-9, with higher scores indicating greater preference”.
Line 203: I struggled a bit with the discussion. One thing that seems to be missing is citing other research studies relevant to your findings on improving the flavor / acceptability of vegetables and promoting change to increase vegetable intake.
Perhaps start the discussion by restating your objective. It might be easier to see that a logical discussion, based upon your methods/results, would be to discuss acceptability (section 4.1) and strategies to increase vegetable intake (section 4.2), and then move onto Implications for Practice (section 4.3) and/or Future Investigation (section 4.4) where you discuss your thoughts on accessibility, availability, sustainability, scalability, and key stakeholders. Much of your discussion is supposition that extends beyond the means of your findings. But it certainly is relevant from an Implication (or Considerations) for Practice perspective.
Response: Thank you for your suggestions. The current structure of the discussion is intended to address topics directly related to the study findings before transitioning into broader implications and future directions. In response to your feedback, we have added a restatement of the study objective at the beginning of the discussion and incorporated citations to strengthen the context.
Line 204: Your study didn't assess accessibility or availability. Your objective was to observe that adding herbs and spice would lead to increased vegetable intake in SM.
Response: Thank you for your feedback. As part of the intervention design, we applied lessons learned from Phase 1 (D’Adamo CR, Troncoso MR, Piedrahita G, Messing J, Scott JM. Spices and Herbs Increase Vegetable Palatability Among Military Service Members. Military Medicine. 2024:usae367. doi:10.1093/milmed/usae367) and addressed previously identified barriers to healthy eating among Service Members. These strategies were intended to meet Service Members where they are, with a focus on improving accessibility and availability of vegetables. We believe these improvements likely contributed to the strong survey response rates and the high levels of vegetable consumption and likeability observed across both preparation types.
Line 205: Do you have a citation for studies with high school students?
Response: Yes, we have added citations to line 205.
Line 208: Consider adding "insignificant" since even though small differences, they are not significant and could be different by chance.
Response: Thank you for the suggestion, to clarify that these are small differences, we have added the word insignificant. Please see line 218.
Lines 212-213: Do you have a citation for this presumption? What about the consideration that many of your participants are likely working in the healthcare field NSA Bethesda is not like a regular military installation) and may be more likely to eat vegetables for health than the majority of other military occupations?
Response: We have added a citation for the Phase 1 study. Regarding the assumption that military personnel in healthcare roles may have different dietary patterns compared to other military occupational specialties (MOS), we are unable to substantiate this claim with existing data.
Lines 229-231: This seems more appropriate if moved up to the last paragraph (end of line 228).
Response: Thank you for your suggestion, instead of lines 229-231 being its own section, we have aligned it to be in the same paragraph as 228.
Line 235: Can you say "making it more likely" if you had no significant increase in consumption. Consider instead "promoting healthier food choice..." or "enabling…".
Response: Thank you for your suggestion, we have used “promoting healthier food choice” instead.
Line 237: I'm not sure this is a valid statement. Military accommodations depend on the rank of the SM, installation, type of duty assignment, etc. Some SMs have mini-refrigerators in their rooms or share a refrigerator in a community kitchen in the barracks. Consider revising to remove "ample".
Response: We have reworded to remove “ample”.
Lines 246-255: Was stakeholder engagement part of the prior paper focused on Phase 1? There is nothing describing the who and what of stakeholder engagement in the methods or results. Perhaps this discussion is really part of Implications for practice. Not a discussion of how your results fit into the current body of knowledge relative to your objective - to examine if herbs and spice will increase vegetable intake in SM.
Response: Thank you for this helpful suggestion. We agree that the discussion of stakeholder engagement is more appropriate in the “Implications for Practice” section rather than in the main discussion of results. We have moved this content accordingly to better align with the study’s objective and to maintain focus on how our findings contribute to the existing body of knowledge.
Line 256: You didn’t examine sustainability or scalability – this really belongs with Considerations (or Implications) for Practice or within a new section for Recommendations for Military Stakeholders.
Response: Thank you for your comment. While we did not directly evaluate sustainability and scalability in this study, we believe that these are important considerations for the success and broader implementation of the intervention.
Line 269: Citation for MREs?
Response: Thank you for your comment. A formal citation is not required in this brief context; however, for additional information, readers can refer to Army Regulation 40-25 (AR 40-25), which outlines nutritional standards and authorized operational rations.
Lines 269-270: Is there a citation to confirm "can be easily incorporated". The US Army Natick Research, Development and Engineering Center is responsible for development and testing of MREs and other operational rations. They would likely be involved in development and likely not an easy incorporation.
Response: Thank you for your comment. Our intention was not to minimize the complexity of the research and development process involved in producing MREs. Rather, we aimed to convey that the incorporation of herbs and spices is a common culinary practice that, if deemed feasible, could potentially enhance palatability within existing ration development frameworks.
Line 275: You could make this section solely focused towards future investigation if moving the accessibility, availability, sustainability, and stakeholder content to a Considerations for Practice section.
Response: Thank you for your suggestion. We have moved the stakeholder engagement content to the “Implications for Practice” section to better align with its focus, and have refined the discussion section to emphasize directions for future research.
Line 289: Need to be careful with interpretation of results. You already had high acceptability and likability scores at baseline with no significant increase in intake with addition of spice and herbs.
Response: Thank you for your suggestion.
Line 291: consider adding "in a larger sample size" after "periods".
Response: Thank you for your suggestion, we have added “in a larger sample size” to clarify this statement.
Line 306: Typically, strength and limitations focus on important characteristics of the study design. One strength you are missing is the phased approach with consistency in meal prep., distribution, and data collection methods to control for bias.
Response: Thank you for bringing this strength to our attention. We have added “Another strength of the study was its robust design, which included a phased implementation approach, standardized meal preparation and distribution, and consistent data collection procedures across all conditions. These elements minimized bias, improved internal validity, and supported reliable comparisons between spiced and plain vegetable options in an operational military setting.” to the strengths and limitations section. Please see lines 328-333.
Line 308: Did you collect data about the barriers? You can't really state it directly addressed the barriers to vegetable intake, unless citing studies that support that presupposition.
Response: We explored common barriers to vegetable intake during the initial phase of this study.
Line 314: Did you report on any of these interactions? Or was this part of Phase 1 cited elsewhere? You talk about the value of having stakeholder input in the section starting on line 246, but I really didn't see any of this highlighted in the methods or results.
Response: Thank you for your comment. The stakeholder engagement described was conducted during Phase 1 of this project, which is cited in the current manuscript. While we did not report those interactions in detail here, the lessons learned from stakeholder input in Phase 1 and previously published research with military service members informed key aspects of this intervention’s design and implementation.
Line 320: as mentioned earlier it is not clear to the reader what your actual sample size was and presented no demographic characteristics about the participants.
Response: We have clarified the sample sizes in the Methods section. Demographic information was not collected from participants as part of this study.
Line 322: This definitely limits generalizability. It is appropriate to remove "may". Also, consider that your population at NSAB is comprised of many SM working in the healthcare field, and may be more inclined to consume more vegetables, not to mention the social desirability of eating more vegetables when participating in a study.
Response: We have removed “may” and added “Furthermore, many Service Members at NSAB work in healthcare-related fields, which could contribute to greater baseline interest in nutrition and vegetable consumption compared to the general SM population. This, along with potential social desirability bias inherent in self-reporting dietary behavior during a nutrition-focused study, may have led to higher intake than would be expected in a less health-oriented population.” to address SM working in the healthcare field.
Line 332: One could consider the SmartIntake Tech a strength compared to the most common data collection method of self-reported intake. Most tools have their strengths and limitations. You could consider including it as a strength as well.
Response: Thank you for pointing out this additional strength, we have included it to read “While the reliance on participant-uploaded data introduces potential issues, such as incomplete or inconsistent submissions, the use of SmartIntake® Technology is also a notable strength of the study.” Please see lines 347-349.
Line 346-347: You just stated a limitation is that convenience, cost, and availability were not explored in as much depth (lines 339-340). And as mentioned earlier, you really only explored the acceptability of vegetables options and intake of plain vs spiced vegetables. Your methods and results do not support including accessibility, availability, or stakeholders in your summary - except to provide a recommendation for future examination. Instead consider focusing on the elevated vegetable intake regardless of spice / herbs, likely due to chef-inspired recipes and your small, specific population. And include that future research is warranted in a larger, more diverse sample of SM to examine if intake between plain and spiced versions was truly by chance considering the plain vegetable intake was higher than expected.
Response: Thank you for your comment. While our study directly measured the acceptability of the intervention, we acknowledge that accessibility and availability are crucial factors influencing vegetable intake. Although not a primary research objective, these elements were indirectly explored through the implementation of the meal kits. We have therefore removed accessibility and availability from the conclusion to accurately reflect the scope of our findings. We appreciate your feedback pointing out this distinction.
Round 2
Reviewer 1 Report (Previous Reviewer 2)
Comments and Suggestions for Authors
The authors have a completely legitimate right to disagree with the reviewer, but this is the first time I have encountered such study in Nutrients. Two tables and a conclusion that emerges even without conducting an experiment, which is "spicy vegetables have greater sensory acceptability and were consumed slightly more".
I believe that this is a valuable result and an important part of a project, just as I believe that for some other journal (not Nutrients), it is enough to examine preferences and sensory characteristics for two vegetables.
In general, there are no shortcomings, but this is exclusively a professional paper in which, in addition to the mean and standard deviation (which are measures of central tendency and dispersion for quantitative, not qualitative data), the p-value was also calculated.
I leave it entirely to the editor to decide what to do with the paper.
This manuscript is a resubmission of an earlier submission. The following is a list of the peer review reports and author responses from that submission.
Round 1
Reviewer 1 Report
Comments and Suggestions for Authors
I read with interest the study aimed to explore the potential of convenience- and sensory-oriented intervention using spices and herbs in heat and serve meal kits to increase vegetable consumption in military settings (MS). The study is an important voice in the discussion on the possibility of improving the quality of diet in SM. The introduction, purpose of the study, and results were well-presented.
The discussion was written correctly. Refers to the results obtained. There are few references to the literature on the topic, perhaps due to the lack of data focusing on vegetable consumption among SM. The Study Limitation, as well as the Future Directions and Potential Implications were taken into account.
I have presented my suggestions below:
Point 1. Page 2, line 46-48 –The authors wrote that "Prior studies have identified barriers to adopting healthier eating habits including limited access to nutritious foods, high costs, military culture, and the prevalence of more convenient fast food option" and no reference was provided. Please provide literature sources.Point 2.
I suggest that information about the 2-phase research model be included in Study design.
Point 3.
Intervention. Was the preparation method (apart from the differences in the use of spices) the same for "plain" and "spicy" vegetables? Is it possible that the final result was influenced by other factors (e.g. aesthetics) apart from the taste)?Point 4.
Section 3.1 is missing. Error in numbering.Point 5.
Were the participants of phase I, the same people who took part in phase II?
The correlation between the subjective assessment of modifiable barriers to vegetable intake and actual, objective vegetable consumption is worth checking.
Author Response
Point 1. Page 2, line 46-48 –The authors wrote that "Prior studies have identified barriers to adopting healthier eating habits including limited access to nutritious foods, high costs, military culture, and the prevalence of more convenient fast food option" and no reference was provided. Please provide literature sources.
Response: Thank you for your comment. We have included 2 references (Troncoso et al, Chukwura et al) to support this statement. Please refer to lines 46-48 for changes.
Point 2.
I suggest that information about the 2-phase research model be included in Study design.
Response: Thank you for your suggestion, we are in agreement with you and have included information about the 2-phase research model in the study design. We added “The study was conducted in two phases. Phase I involved a comprehensive sensory evaluation—assessing taste, appearance, aroma, and texture —while Phase II focused on evaluating vegetable intake was designed based on the outcomes of the Phase I”. We also provided a citation to the publication of our Phase I manuscript (PMID: 39078749). Please see lines 85-88.
Point 3.
Intervention. Was the preparation method (apart from the differences in the use of spices) the same for "plain" and "spicy" vegetables? Is it possible that the final result was influenced by other factors (e.g. aesthetics) apart from the taste)?
Response: The preparation method was kept consistent for both spiced and plain versions. However, intake is inherently affected by a range of sensory elements beyond taste, such as appearance, aroma, and texture, which may have contributed to the observed consumption patterns. To clarify, we have added “While consistent preparation reduces potential confounding, factors beyond taste, including appearance, aroma, and texture, may also influence intake and preference.” to this section. Please refer to lines 121-123.
Point 4.
Section 3.1 is missing. Error in numbering.
Response: Thank you for catching this error. We have updated 3.2 to 3.1.
Point 5.
Were the participants of phase I, the same people who took part in phase II?
Response: Yes, the participants for both phases were recruited from the same group of service members at Naval Station Activity Bethesda (NSAB). However, it is possible that some individuals participated only in Phase I or only in Phase II.

Reviewer 2 Report
Comments and Suggestions for Authors
The idea of the manuscript; explore the potential of a convenience- and sensory-oriented intervention using spices and herbs in heat and serve meal kits to increase vegetable consumption in military settings, is of exceptional importance, however, there are many ambiguities such as:
- clearly defined objective of the work
- it is not known how many participants were included in the evaluation and a flowchart of research activities would be of exceptional help in understanding the implementation of the research
- it talks about sensory, evaluations, etc. and the results include tables that list the average cups consumed and Mean likeability scores and that's it.-for a journal as Nutrients?
Sincerely
Author Response
- clearly defined objective of the work
Response: Thank you for your comment. We have defined the objective in the abstract: “This study aimed to evaluate whether heat and serve meal kits with spices and herbs could increase vegetable intake and liking among active-duty SM, by simultaneously targeting barriers to healthy eating and modifiable sensory factors.” (line 17-18). In addition, we have also stated the objective in the introduction: “Therefore, the purpose of this study was to evaluate whether the addition of spices and herbs (spiced) to vegetables provided as part of a heat-and-serve meal kit meals served to active-duty SM will increase vegetable intake (primary outcome) and vegetable liking (secondary outcome) compared to plain vegetables” (lines 76-79).
- it is not known how many participants were included in the evaluation and a flowchart of research activities would be of exceptional help in understanding the implementation of the research
Response: Thank you for your suggestion. We note that 400 meals were distributed; however, we cannot confirm the exact number of unique participants in the evaluation, as the data were collected anonymously, and individuals were allowed to participate multiple times, permitted to receive one meal per vegetable. While a flow chart could enhance the visual presentation of the research process, it may be redundant, as the research activities are already described in detail in lines 105-149. However, if you feel that it is imperative that we include a flow chart of some sort in light of the anonymized and overlapping enrollment of participants for the different meals that were served, we can do so.
- it talks about sensory, evaluations, etc. and the results include tables that list the average cups consumed and Mean likeability scores and that's it.-for a journal as Nutrients?
Response: Thank you for your insights. In the introduction of the revised manuscript, we briefly touch upon and have cited the publication describing Phase I of this research (PMID: 39078749), which included monadic sensory testing of flavor, appearance, aroma, and texture (lines 69-73). In order to connect the previous phase’s results about sensory evaluation to our current Phase II findings, we have added a summary of the findings to the results section. The added section reads “The preliminary phase of this research identified appearance, preparation style, and taste as the main barriers to vegetable intake in military dining facilities, with sensory testing showing a preference for spiced vegetables over traditional preparations in terms of appeal, flavor, and aroma (P < .03).” We have reworded the next sentence to transition into phase II results. This section reads “Building on the findings from Phase I, Phase II further explored these barriers, distributing 400 meals to gather data on vegetable consumption and liking ratings for both spiced and plain preparations of four vegetables: broccoli, carrots, cauliflower, and kale.” Please refer to lines 162-168 for these changes.
Round 2
Reviewer 2 Report
Comments and Suggestions for Authors
I'm sorry, but the comments I made were not "minor" so that through additions in L46-48; L85-88; L121-123 and L162-168 could be accepted for a scientific article.
The aforementioned changes did not clearly link the goal (hypothesis), results, discussion and conclusions, therefore I still suggest not to publish the work (at least not in the journal Nutrients).
Author Response
RESPONSES TO REVIEWER 2
Comment 1: I'm sorry, but the comments I made were not "minor" so that through additions in L46-48; L85-88; L121-123 and L162-168 could be accepted for a scientific article.
Response: Thank you for your feedback. We appreciate your continued engagement with our manuscript and the opportunity to further clarify and strengthen our submission. We fully acknowledge that the issues raised were substantial and important for ensuring scientific rigor. In response, we carefully addressed each of your points with detailed explanations and additions throughout the revised manuscript, particularly in the areas you identified (e.g., L46–48, L85–88, L121–123, and L162–168).
Specifically, we provided more extensive context from Phase I, including differences in preference ratings and methodological refinements made for Phase II. We also addressed variability in response numbers and participant feasibility constraints, included references and validation statistics for the intake measurement method (Remote Food Photography Method), clarified the multiple meal components and added details on the consistency across meal preparations.
Comment 2: The aforementioned changes did not clearly link the goal (hypothesis), results, discussion and conclusions, therefore I still suggest not to publish the work (at least not in the journal Nutrients).
Response: Thank you for your candid feedback regarding the connection between the study’s goal, results, discussion, and conclusions. We appreciate the opportunity to clarify the logical flow of our manuscript and have made additional revisions to enhance this alignment.
Our primary hypothesis was that providing heat-and-serve vegetable meal kits enhanced with spices and herbs would improve both vegetable intake and liking among military Service Members by addressing two major barriers: convenience/accessibility and taste preference. This hypothesis is clearly stated in the revised introduction and background sections, and is reiterated in the discussion and conclusion for clarity and consistency. In the results, we report that vegetable intake was very high overall (87%) and that liking scores and intake did not significantly differ between the plain and spiced preparations. While the lack of a significant difference between versions did not support our hypothesis about the added benefit of spices and herbs, the findings did support our broader goal: convenient access to meal kits substantially improved overall vegetable intake in a population with historically low consumption. This is a key implication, which we have emphasized more directly in the discussion and conclusion.
To address your comment more directly, we have strengthened the connection between our:
● Stated hypothesis (meal kits + spices/herbs will improve intake and liking),
● Results (very high intake for both conditions),
● Discussion (impact of convenience and high acceptance),
● Conclusion (highlighting the success of convenient meal kits in overcoming access
barriers, and refining future research focus).
We hope these revisions address your concern and clarify how the study meaningfully
contributes to the literature on improving diet quality in military settings.